# An *in silico* framework for the rational design of vaginal probiotic therapy

**Christina Y. Lee[‡], Sina Bonakdar[‡], Kelly B. Arnold** *

Department of Biomedical Engineering, University of Michigan, Ann Arbor, Michigan, United States of America

‡ These authors share first authorship on this work.
* kbarnold@umich.edu

**Data Availability Statement:** All code and data used in this study are available at: https://github.com/chyylee/CST_Probiotic.

## Abstract

Bacterial vaginosis (**BV**) is a common condition characterized by a shift in vaginal microbiome composition that is linked to negative reproductive outcomes and increased susceptibility to sexually transmitted infections. Despite the commonality of BV, standard-of-care antibiotics provide limited control of recurrent BV episodes and development of new biotherapies is limited by the lack of controlled models needed to evaluate new dosing and treatment regimens. Here, we develop an *in silico* framework to evaluate selection criteria for potential probiotic strains, test adjunctive therapy with antibiotics, and alternative dosing strategies. This computational framework highlighted the importance of resident microbial species on the efficacy of hypothetical probiotic strains, identifying specific interaction parameters between resident non-optimal anaerobic bacteria (nAB) and *Lactobacillus* spp. with candidate probiotic strains as a necessary selection criterion. Model predictions were able to replicate results from a recent phase 2b clinical trial for the live biotherapeutic product, Lactin-V, demonstrating the relevance of the *in silico* platform. Results from the computational model support that the probiotic strain in Lactin-V requires adjunctive antibiotic therapy to be effective, and that increasing the dosing frequency of the probiotic could have a moderate impact on BV recurrence at 12 and 24 weeks. Altogether, this framework could provide evidence for the rational selection of probiotic strains and help optimize dosing frequency or adjunctive therapies.

## Author summary

Bacterial vaginosis (**BV**) is a common condition that affects approximately 30% of reproductive-age women and is associated with pre-term birth, pelvic inflammatory disease, and increased susceptibility to infection. Standard antibiotic regimens used to treat BV have long-term cure rates of less than 50%, and probiotics have been proposed as an alternative therapeutic option. Here we present a new computational framework to evaluate vaginal microbiome communities in the presence of a probiotic species with the goal of inferring probiotic characteristics and treatment regimens that best promote transitions to optimal healthy states across heterogeneous patient populations. Results highlight the critical importance of selecting probiotic strains that have commensal relationships with

**Funding:** This work is supported by NIH T32GM141746 and funds from the University of Michigan to KBA. Funders did not play any role in study design, data collection and analysis, decision to publish, or preparation of the manuscript.

**Competing interests:** The authors have declared that no competing interests exist.

existing community members, which is different than the traditional emphasis on selecting strains that inhibit BV-associated species. We also demonstrate how this framework can be used for hypothesis-testing of combined antibiotic and probiotic treatment regimens, with the overall goal of improving future rational design of therapies to combat BV.

## Introduction

The vaginal microbiome (**VMB**) is critical to female reproductive health, with the composition of the VMB associated with benefits such as decreased risk for sexually transmitted infections [1–5], pelvic inflammatory disease [6–8], and pre-term birth [9–11]. Clinical observations support that the VMB is typically dominated by a single *Lactobacillus* spp. or exist in a polymicrobial state where no one species dominates the community [12,13]. These compositional states are referred to as community state types (CSTs) and are identified through an established nearest centroid classifier called VALENCIA [14].

The most health-associated compositions are associated with a lack of diversity and *Lactobacillus* spp. dominance, particularly by *L. crispatus*, *L. gasseri*, and *L. jensenii* (CSTs, CST -I, CST -II, CST -V). These species are able to facilitate the acidification of the vaginal microenvironment through the production of D, L-lactic acid and are also reported to produce hydrogen peroxide, which decrease susceptibility to sexually transmitted infections and inhibit the growth of pathogens and other non-optimal species [1,15–18]. Additionally, *Lactobacillus* spp. can produce bacteriocins and surfactants which can also prevent colonization of non-optimal or pathogenic species [19]. A fourth *Lactobacillus* sp., *L. iners*, is linked to CST -III and lacks the ability to produce compounds most associated with the inhibition of pathogens ($H_2O_2$, D lactic acid) [17,20]. While *L. iners* is still considered healthy and does not promote adverse clinical symptoms, it has been associated with shifts to a non-optimal composition characterized by an overgrowth of facultative and obligate anaerobes known as bacterial vaginosis (BV) [1,17,21]. Two of the most commonly associated bacteria associated with BV are *Gardnerella vaginalis* and *Prevotella* spp., which can exist at low levels even in *Lactobacillus* spp. dominated communities [12]. However, typically no one species dominates the community during episodes with BV and herein this diverse community that lacks dominance by a species considered to be optimal (*Lactobacillus* spp.) is referred to as a non-optimal bacteria (nAB) dominant group.

BV is a common condition, affecting approximately 30% of reproductive-age women resulting in abnormal vaginal discharge and odor, discomfort, and higher risk for the adverse reproductive outcomes. Despite the commonality of BV, treatment outcomes with standard antibiotic regimens (nitroimidazoles, primarily metronidazole, or clindamycin) remain suboptimal, with short-term cure rates around 80% [22] and long-term (6–12 months) cure rates at less than 50% [23]. Thus, alternative methods for long-term resolution of BV are needed. Several alternative strategies for treating recurrent and repeated episodes of BV have been evaluated based on empirical observations, which include extended first-line antimicrobial regimens, combinatorial first-line regimens, therapies targeted at biofilm removal (boric acid, TOL-463), pH lowering agents (lactic acid), and probiotics. Women with persistent or recurrent BV prescribed twice weekly doses of metronidazole for three months reduced recurrences during therapy, but once discontinued had repeat episodes of BV [24,25]. There is limited data on the use of boric acid in conjunction with long-term suppressive antimicrobial regimens, with one study supporting higher cure rates at 12, 16 and 28 weeks, but by week 36 was less than 50% [26]. pH lowering agents like lactic acid have also been studied, but have not shown

ability to significantly impact VMB composition and are not recommended by any guidelines [27]. Many randomized control trials have tried to support the use of probiotics for the treatment of BV, with mixed results [28–33]. One recent trial of *L. crispatus* CVT-05 (Lactin-V) showed promise for reducing the recurrence of BV at 12 weeks when compared to placebo, but is not yet cleared by the FDA or commercially available [29]. The study and use of probiotics for the treatment of BV has been limited by inconsistent probiotic characteristics (vaginal vs intestinal species, vaginal strains), routes of administration (oral vs vaginal), and dosing strategies (frequency and duration) [34]. Methodical selection of probiotic characteristics and dosing regimens could greatly improve efforts to develop a probiotic or live biotherapeutic products that can resolve recurrent BV.

Here, we develop model that can systematically test probiotic characteristics and dosing strategies against a variety of *in silico* BV communities that reflect heterogeneous patient populations. The model reveals that resident community members can have a significant impact on probiotic efficacy, particularly highlighting that any antagonistic interaction of non-optimal anaerobic bacteria (nAB) on the probiotic strain can drastically decrease probiotic success at re-orienting communities to a *Lactobacillus* spp. dominant state. Additionally, we observe that the relationship between resident *Lactobacillus* spp. with probiotic can impact whether the post-treatment communities will be dominated by optimal *Lactobacillus* (oLB), *L. iners*, or the probiotic species. Lastly, the modeling framework was evaluated in the context of regimen reported for Lactin-V in a phase 2b clinical trial demonstrating the model can replicate clinical observations. Overall, these results highlight the importance of characterizing probiotic strains in co-culture with endogenous VMB and suggest personalized differences in microbial characteristics can help explain variability efficacy observed clinically.

## Results

### Simulated probiotic strains result in variable response types across a virtual population

To simulate BV communities, we used a three-community state type (CST) Ordinary Differential Equations (ODE) based model to represent core VMB compositional types: optimal *Lactobacillus* (*L. crispatus*, *L. jensenii*, or *L. gasseri*) dominant (oLB; CST -I/II/V), *L. iners* (Li; CST -III), and non-optimal anaerobic bacteria (nAB, associated with BV; CST -IV). A probiotic (P) is also included in the community in the form of a fourth representative species that interacts with other species in the community (Fig 1A). The model captures the growth characteristics and interspecies interaction terms between each community group as well as how the community and the probiotic interact using generalized Lotka-Volterra equations (Fig 1A) [35–38]. Within this model, the inter-species interaction terms quantify the impact of one species on another. The sign of each term or whether the interaction is not significant (zero value) can be related to ecological terms such as commensalism (+/0 or 0/+), symbiosis (+/+), amensalism (-/0 or 0/-), competition (-/-), or parasitism (+/- or -/+) [39]. For example, a commensal relationship between two species may exist due to the production of a metabolite that increases the growth of the other species (+/0) [40]. In contrast, a common example of an amensal or competitive interaction can be observed between *Lactobacillus* spp. which produce lactic acid or other inhibitory compounds against nAB while only some nAB may produce compounds like biogenic amines that inhibit *Lactobacillus* spp. growth (-/- or -/0 interactions) [16,41]. Many of these growth and interaction terms vary in microbial strains and across individuals. While probiotic strains are typically *Lactobacillus* spp., this general framework also enabled the assessment of different probiotic strains and species by varying probiotic interactions parameters ($\alpha$).

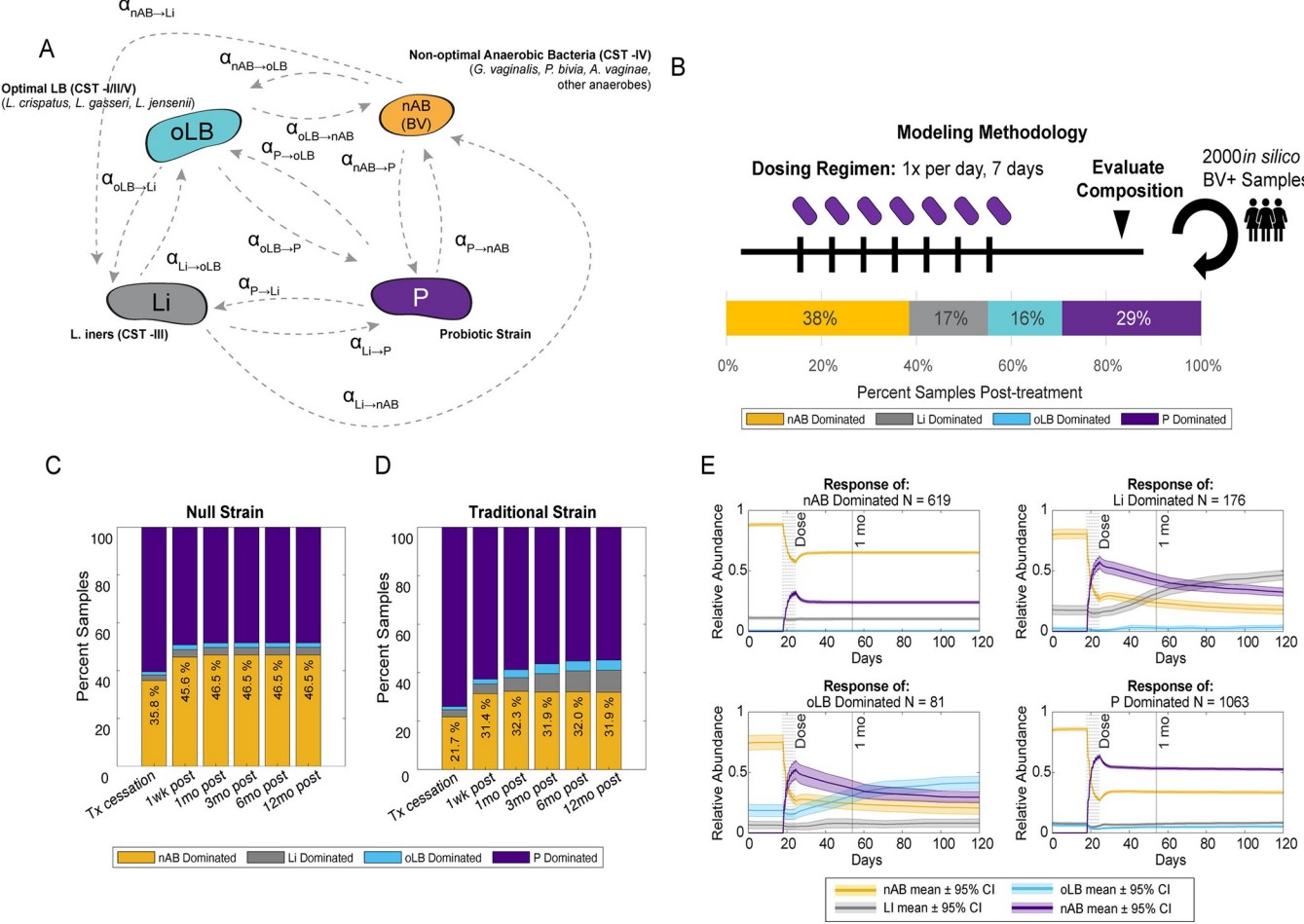

**Fig 1. Overview of Probiotic Strain Modeling.** (A) Schematic of parameters associated with probiotic–resident community interactions. (B) Base probiotic dosing regimen modeling. Unless otherwise specified, probiotic dosing occurs once daily for seven days across 2000 virtual BV+ patient samples generated in a previously published manuscript [39]. At several time points the dominating species is assessed, which can result in a mix of nAB-, Li-, oLB-, or Probiotic-dominant communities for the same strain of probiotic. (C) Results for a "null" strain probiotic that has a moderate growth rate (0.5 d⁻¹) and negligible interspecies interactions, the stacked bar graph represents the frequency of communities in one of four states: nAB-dominant, Li-Dominant, oLB-Dominant, or Probiotic- dominant at time points therapy cessation, 1 wk, 1 mo, 3mo, 6mo and 12mo post. (D) Example of a conceptually traditional probiotic strain (null parameters with $\alpha_{P->nAB}$ = -0.010 density⁻¹d⁻¹). (E) Example time series relative abundance responses of the *in silico* BV+ patients. Each plot is the average ± 99% confidence interval of the relative abundance for each species of *in silico* BV+ patients that exhibited a set response type of nAB-dominant (top left), Li-dominant (top right), oLB-dominant (bottom left), and P-dominant (bottom right) by simulating traditional strain.

In order to capture heterogeneity observed in microbial communities we used virtual patient populations generated in a previously published paper (Lee et al., 2023)[42]. Briefly, this methodology generated base virtual population using Latin Hypercube Sampling of defined parameter distributions from literature (30,000 parameter sets, S1 Table, S1 Fig). From the base virtual population of Latin Hypercube Sampling- generated parameter sets, 2,000 virtual patients were selected to replicate a distribution of BV-relevant CST equilibrium behaviors observed in the HMP cohort clinically [42]. This population was used to test differ- ent probiotic characteristics and regimens. BV-relevant CST equilibrium behaviors were defined as virtual patients that were analytically predicted to be nAB- dominant at steady-state [42,43]. For proof of concept, a simple, 7-day regimen of probiotic was simulated on each vir- tual patient, and the impact on community composition was evaluated at several time points from therapy cessation (therapy cessation, 1 month, 3 months, 6 months, 12 months; Fig 1B).

The impact of the probiotic on community composition was defined using VALENCIA, a classification algorithm that is used clinically to assign CSTs at a set evaluation time point [14]. These classifications included nAB-dominant (probiotic failure rate) as well as dominant by *Lactobacillus* spp. (probiotic efficacy rate): Li-dominant, oLB-dominant, or Probiotic (P)-dominant.

Current characterization or interactions between probiotic bacteria strains and bacteria present in the vagina primarily focus on the impact of the probiotic on nAB associated with BV. There is limited data on how probiotic strains impact endogenous *Lactobacillus* spp. or how endogenous community members impact probiotic strains. For simplicity, a hypothetical null probiotic strain and a traditional probiotic strain were created to explore the possible response types. The null probiotic strain (control strain which exhibits no impact on the endogenous community and vice versa) was modeled as having negligible interspecies interactions ($\alpha_{nAB \to P}$, $\alpha_{Li \to P}$, $\alpha_{oLB \to P}$, $\alpha_{P \to nAB}$, $\alpha_{P \to Li}$, $\alpha_{P \to oLB}$ = 0.0 density$^{-1}$d$^{-1}$) a moderate growth rate ($k_{grow-P}$ = 0.5 d$^{-1}$), and a moderate self-interaction term ($\alpha_{P \to P}$ = -0.022 density$^{-1}$d$^{-1}$). The range of growth and self-interaction terms were characterized from *in vitro* experiments [37] (S1 Table). The growth rate ranges from 0.1 to 1, with 0.5 considered moderate. For the self-interaction parameters, the average value is -0.022, derived from the provided range of -0.004 to -0.04 in S1 Table. Resulting model predictions indicated that the null probiotic strain was predicted to have primarily P-dominant or nAB-dominant response types at each evaluation point, with a maximal failure rate of 46.5% at 12 months (Fig 1C). A second hypothetical strain, designed as a proof of concept example with traditional considerations of ensuring the probiotic strain could inhibit nAB growth was simulated using the null probiotic strain parameters and $\alpha_{P \to nAB}$ set to -0.01 density$^{-1}$d$^{-1}$ (Fig 1D). Steady-state was achieved by 1 month for the proportion of nAB-dominant response types after treatment where the traditional strain exhibited a higher frequency of treatment success defined by lower frequency of nAB-dominant virtual patients after treatment compared to the null group. At and after the 1 month evaluation time point, 32% of virtual patients were nAB-dominant for the traditional strain compared to 46.5% for the null strain, which was significantly different (P < 1E-6, $\chi^2$-test for frequency of nAB-dominant responses).

To demonstrate how community composition changes during and after the probiotic therapy, abundance-time profiles were plotted for each response type (Fig 1E). For nAB-dominant and P-dominant response groups, nAB relative abundance was lowest on the last day of therapy and then re-equilibrated to a higher abundance. For responses where endogenous *Lactobacillus* spp. were the most abundant post-treatment (Li- or oLB-dominant response types), nAB populations continued to decline over time alongside the probiotic population indicating the presence of probiotic benefited the endogenous *Lactobacillus* spp. populations enough to alter community composition. Altogether these results demonstrate the utility of testing a probiotic strain and regimen across a heterogenous simulated population, as even the same probiotic strain can induce variable responses across patients.

### Sensitivity analyses reveal alternative probiotic characteristics that can improve probiotic strain efficacy

To determine which parameters are most associated with improving probiotic efficacy (frequency of Li-, oLB-, P-dominant states post-treatment), local sensitivity analyses were conducted for all parameters describing probiotic strain characteristics (Fig 2). The level of sensitivity for each parameter and response type, was quantified using a previously published method, which normalizes the changes in outcome by the change in parameter value [44]. Notably, among the interaction parameters, the most sensitive parameters for the nAB-

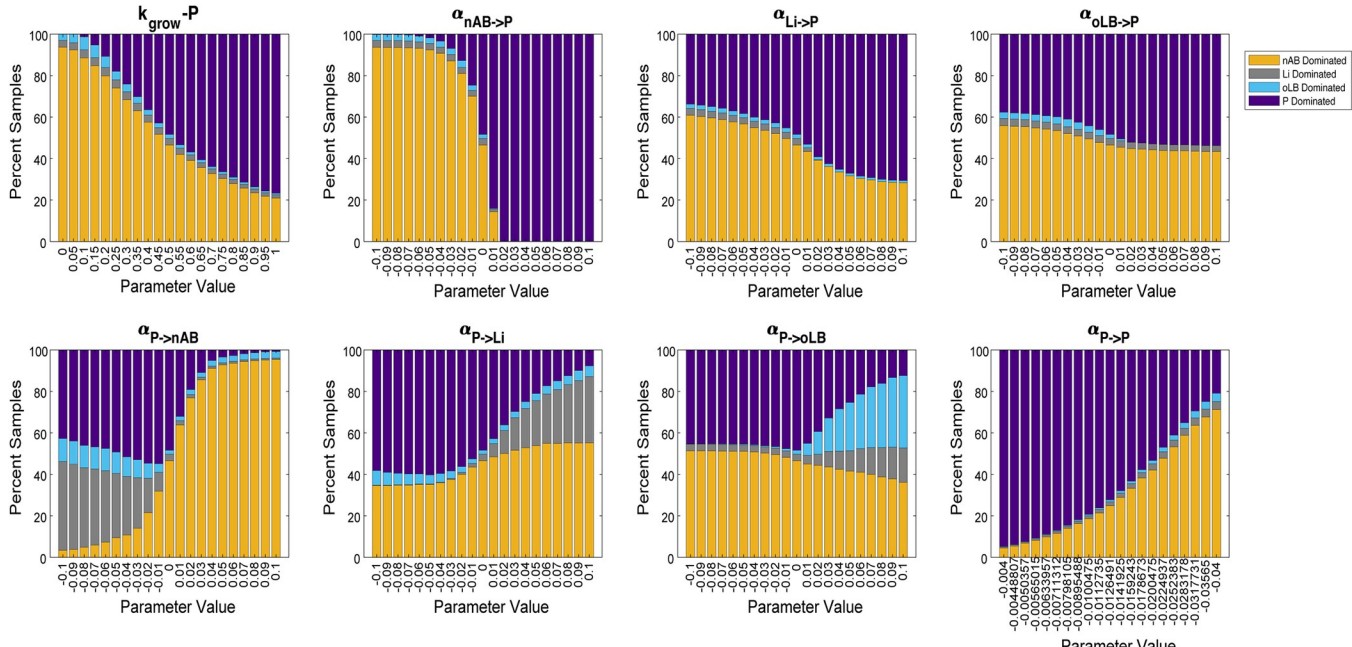

**Fig 2. Local sensitivity analysis of probiotic characteristics.** One-at-a-time parameter perturbation analysis for probiotic characteristics. The proportion of the 2,000 virtual BV+ communities that exhibited each response type at 12 months post therapy is plotted on the y-axis. The x-axis represents the parameter values where for $k_{grow-P}$ the null value is 0.5 $d^{-1}density^{-1}$, interaction terms the null value is 0 $d^{-1}density^{-1}$, and for the self-interaction term is -0.0220 $d^{-1}density^{-1}$.Each parameter was perturbed within its own minimum and maximum value range. The sensitivity of each parameter for all response types was quantified using a normalized sensitivity metric [44] and is provided in Tabe S3.

dominant response (probiotic failure) were between nAB and probiotic ($\alpha_{nAB \to P}$ and $\alpha_{P \to nAB}$) with the highest absolute scores (-0.32 and 0.07, respectively; S3 Table). The importance of probiotic inhibition on nAB ($\alpha_{P \to nAB}$) is unsurprising, given that the selection of probiotics is often based on the ability to produce inhibitory compounds (D/L-lactic acid, hydrogen peroxide, or bacteriocins) for nAB [15,19,45–47]. In contrast, the high degree of sensitivity for nAB on probiotic ($\alpha_{nAB \to P}$) is less intuitive and not well characterized *in vitro* or *in vivo*. The pairwise interactions of endogenous *Lactobacillus* (Li and oLB) with the probiotic did not have as strong of an effect on predicted probiotic efficacy ($\alpha_{Li \to P}$, $\alpha_{oLB \to P}$, $\alpha_{P \to Li}$, $\alpha_{P \to oLB}$). However, these interactions could alter which type of *Lactobacillus* spp. dominant state the community assembled to post-treatment ($\alpha_{P \to Li}$, $\alpha_{P \to oLB}$). The ability to dictate which *Lactobacillus* spp. community dominates post-treatment could be useful, as recent studies have started to design therapies to inhibit *L. iners* populations as a methodology to prevent BV recurrence [48,49].

To analyze the impact of combinatorial changes in probiotic characteristics, a four-parameter sensitivity analysis was used. The four-parameter sensitivity analysis covered the two parameters most sensitive for probiotic efficacy ($\alpha_{nAB \to P}$ and $\alpha_{P \to nAB}$) and for specificity to boost oLB or Li-dominant response types ($\alpha_{P \to Li}$, $\alpha_{P \to oLB}$; S2 Fig). Each parameter was simulated as a -0.01, 0.00, or +0.01 $density^{-1}d^{-1}$ value, for all possible combinations (81 parameter combinations total). Overall, the probiotics with the lowest failure rate (nAB-dominant frequency) exhibited a positive value for $\alpha_{nAB \to P}$. The next important driver was $\alpha_{P \to nAB}$.

Based on the results of the sensitivity analysis, we aimed to identify the strain with the lowest failure rate. We simulated our model with three hypothesized strains, including the best strain given only one parameter is altered: 1p: traditional strain (positive $\alpha_{nAB \to P}$), the best strain given two parameters are altered: 2p: best strain (positive $\alpha_{nAB \to P}$ and negative $\alpha_{P \to nAB}$), and when three parameters are altered the 3p: best strain (positive $\alpha_{nAB \to P}$, negative $\alpha_{P \to nAB}$,

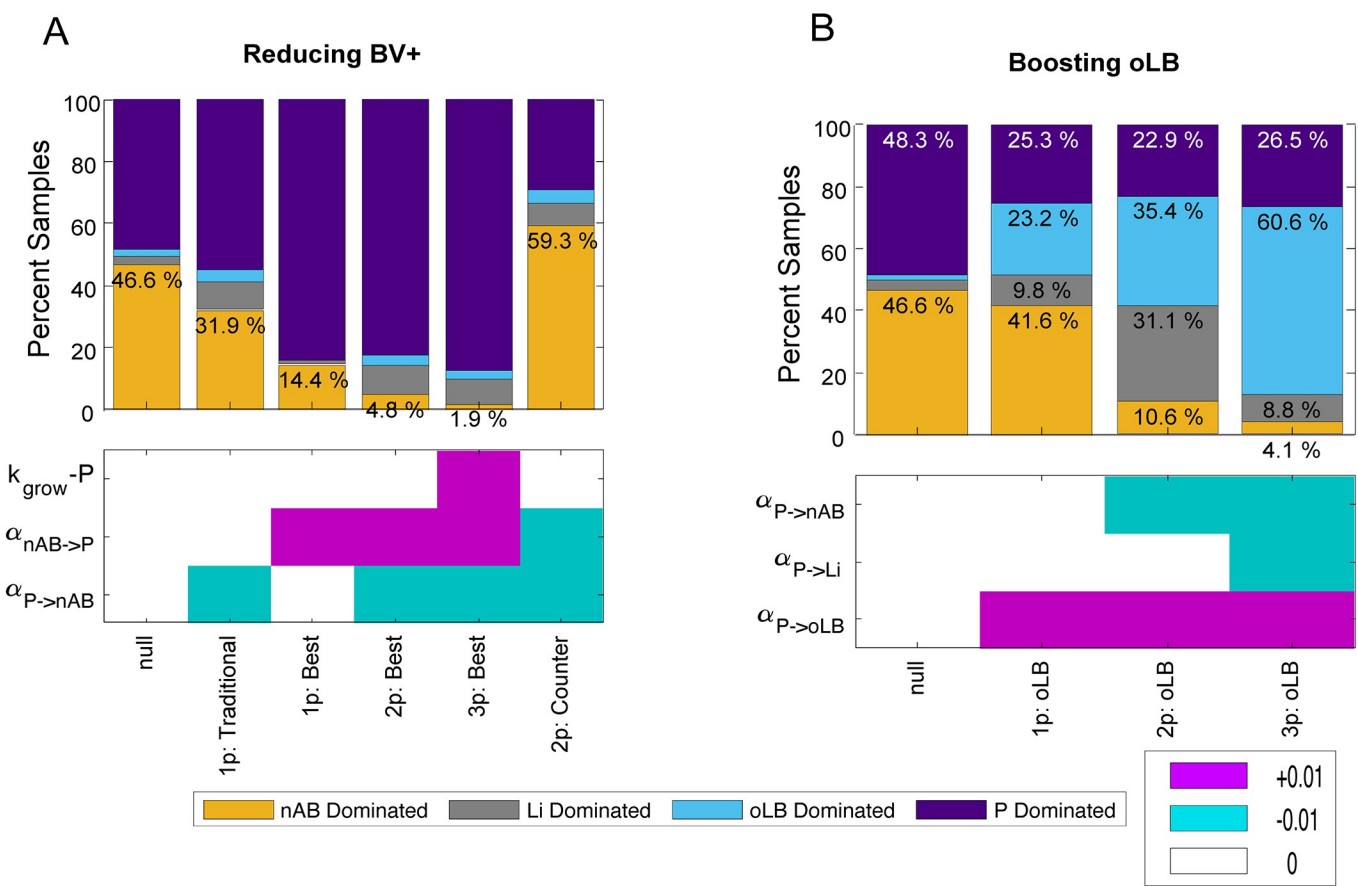

**Fig 3. Selection of Optimal Probiotics.** Based on local sensitivity analysis in Fig 2, hypothetic probiotic strains were selected with 1, 2 and 3 parameter changes to observe how much impact the combination of parameters could have on treatment outcomes. The parameters that were varied for each hypothetic strain are shown in the heatmap below the plots. The heatmap indicates the parameter change and the value of the change: no change (white), +0.01 (pink), -0.01 (teal). (A) Hypothetical strains that would be predicted to improve probiotic efficacy by reducing nAB-dominant response types. (B) Probiotics that boost endogenous oLB populations to increase oLB-dominant response types. Plots indicate the percentage of the 2,000 *in silico* BV+ samples for a given response type. Statistical comparisons were made with chi-square tests and all responses were evaluated at 12 months post treatment cessation.

and positive $K_{grow-P}$). Statistical analysis of selected strains versus the null probiotic strain was used to emphasize the best 1-parameter, 2-parameter, and 3-parameter probiotic designs (Fig 3A). All selected strains were significantly different from the null strain. The traditional strain (1 parameter modification of $\alpha_{P\to nAB}$) had a significantly higher rate of treatment failure than the 1-parameter alteration of $\alpha_{nAB\to P}$ (31.9% vs 14.4%; $P < 1\times10^{-6}$), highlighting the importance of considering the effect of endogenous nAB populations on probiotic efficacy. Combining the two best single-parameter perturbations decreased the treatment failure rate to 4.8%, promoting probiotic growth rates alongside the 2-parameter change (3p-best strain) decreased the failure rate to 1.9% ($P = 6.141\times10^{-7}$). To determine the impact of a competitive interaction, a strain where both $\alpha_{nAB\to P}$ and $\alpha_{P\to nAB}$ are negative was simulated (2p: Counter). When $\alpha_{nAB\to P}$ is negative, it can counteract the effect of a probiotic selected for inhibition of nAB (negative $\alpha_{P\to nAB}$) and perform worse than a probiotic with no interspecies interactions (null strain, 59.3% vs 46.5%; $P = 1.221\times10^{-15}$). Altering the impact of probiotic on nAB, Li, and oLB were observed to impact the selection of Li- and oLB-dominant response types in the 1-parameter sensitivity analysis. Selection of parameter modifications to boost oLB-dominant response types was assessed, increasing the oLB-dominant response type from 2.0%, to 23.2%

with a 1-parameter modification (1p: oLB; +0.05 $\alpha_{P \to oLB}$), 35.4% with a 2-parameter modification (2p: oLB; +0.05 $\alpha_{P \to oLB}$ & -0.05 $\alpha_{P \to nAB}$), and 60.6% with a 3-parameter modification (3p: oLB;+0.05 $\alpha_{P \to oLB}$, -0.05 $\alpha_{P \to Li}$, & -0.05 $\alpha_{P \to nAB}$; Fig 3B). These results suggest that understanding the interaction between nAB and probiotic, as well as the impact probiotic has on endogenous *Lactobacillus* spp. can help tailor desired compositional changes with probiotics.

## Combinatorial regimens can lower BV recurrence rates

Commonly, a course of probiotics is given after treatment with standard antimicrobial therapy [34] (S2 Table). To evaluate the impact of different treatment regimens, the model was used to simulate pre-treatment with antibiotics followed by a short-term probiotic regimen, versus a short-term probiotic only regimen, and antibiotic only regimens (Fig 4A). The antibiotic therapy simulated was a 7-day course of oral metronidazole (**MTZ**) and the probiotic regimen was a short-term (7-day), daily regimen using the traditional strain (null strain with $\alpha_{P \to nAB}$ = -0.01 density$^{-1}$d$^{-1}$). Treatment outcomes were evaluated at the end of treatment, 1 month, 3 months, and 6 months post-treatment cessation (Fig 4A). The antibiotic regimen without probiotic performed the worst at all time points except immediately after treatment cessation, where the antibiotic regimen had a 15% failure rate versus the 21.6% failure rate for the probiotic only regimen (P < 1x10$^{-6}$). At all evaluation time points, the combination pre-treatment antibiotic followed by probiotic was most efficacious. At the later time points, the difference between combination antibiotic with probiotic versus probiotic only regimen decreased with the failure rate within 4% of each other. The failure rate for antibiotic at 3 month and 6 months was nearly double that of the regimens including probiotic, supporting the use of probiotic to reduce repeat episodes of BV.

Probiotic regimens can vary significantly, with some strategies using short-term (daily) regimens, intermittent regimes (weekly), or long-term regimens (treatment over several months). A promising probiotic and associated regimen (Lactin-V) recently underwent phase 2 clinical trial demonstrating significant reduction in BV recurrence and utilized a combination of the aforementioned dosing strategies [29]. The phase 2b trial administered a 5-day regimen of intravaginal MTZ, followed by vaginally administered Lactin-V or placebo. The placebo in the phase 2b study was all components of the Lactin-V formulation except the probiotic strain, *L. crispatus* CTV-05. Dosing of Lactin-V probiotic strain/placebo occurred over 11 weeks, where week 1 was dosed once daily for 4 days, and weeks 2–11 were dosed twice weekly. To demonstrate the model can recapitulate complex dosing strategies and replicate clinical observations, the Lactin-V regimen was simulated across 2,000 virtual patients and the predicted recurrence rates at 12 weeks and 24 weeks were compared to clinically reported frequencies. The placebo (5-day regimen of MTZ followed by the Lactin-V regimen with no probiotic strain simulated) results agreed well between clinical observation and model results at 24 weeks (66.1% versus 71.0%; P = 0.4029), but deviated at the 12 week mark (53.1% versus 70.5%; P = 0.002; Fig 4B). Since the strength of antibiotic is a parameter that exhibits significant variability (5-fold differences in decay rates; S2 Table), a stronger dose was simulated which demonstrated comparable results at both the 12 and 24 week mark for the placebo arm (S3 Fig). Since characteristics of Lactin-V strain, *L.* crispatus CTV-05 are not well characterized in the literature, four different hypothetical strains were benchmarked against the clinical results in the phase 2b clinical trial. Strength of interaction terms were defined relative to the strength of probiotic inhibition of nAB which was fixed to -0.012 d$^{-1}$density$^{-1}$

1. The "traditional" probiotic strain with strong inhibition of P on nAB (simulation positive control, -0.012 d$^{-1}$density$^{-1}$ $\alpha_{P \to nAB}$).

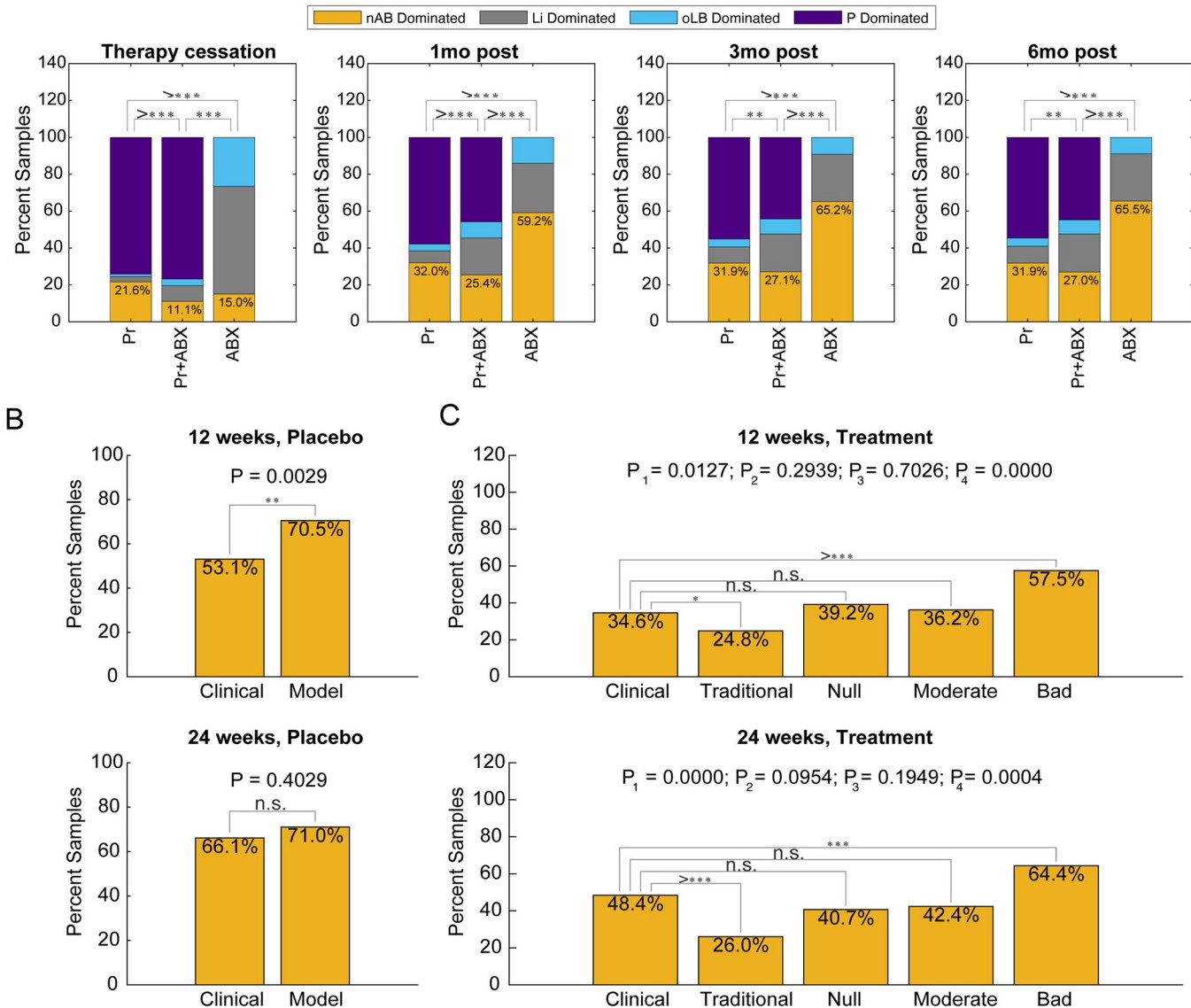

**Fig 4. Adjunctive antimicrobial therapy versus probiotic or antibiotic therapies in isolation.** (A) Assessment of the impact of adjunct antimicrobial therapy (ABX + Pr) versus probiotic only (Pr) or antibiotic only (ABX) BV treatment regimens across four four time points (therapy cessation, 1 month post, 3 months post and 6 months post). The impact of antibiotic was simulated at a moderate magnitude (-2.64 d^-1) that was equivalent to the average decay rate of nAB species with ABX reported by Mayer et al. 2015 [69]. The percent of the 2,000 in silico BV+ subjects that exhibit each response type are reported. (B-C) Comparison of model predictions with Lactin-V trial results at 12 and 24 weeks for (B) placebo (antibiotic treatment only) (C) treatment arm (antibiotic followed by probiotic treatment). For the treatment arm, 4 strains were simulated by the model encompassing a traditionally designed probiotic, null probiotic, moderately/ conservatively designed probiotic, and bad/negative control probiotic.

2. The "null" probiotic strain.

3. A "moderate" strain with strong competition with endogenous *Lactobacillus* spp. vaginal microbiota (3X strength of $\alpha_{P \to nAB}$), weak inhibition of nAB on P (1/2X $\alpha_{P \to nAB}$), and strong inhibition of P on nAB (-0.012 $d^{-1}$density$^{-1}$ $\alpha_{P \to nAB}$).

4. A "bad" strain with strong competition with endogenous *Lactobacillus* spp. (3X strength of $\alpha_{nAB \to P}$), weak inhibition of P on nAB (1/2X traditional strain strength of $\alpha_{P \to nAB}$), and strong antagonism of nAB against the probiotic (1X traditional strain strength of $\alpha_{P \to nAB}$) (simulation negative control).

The traditional probiotic strain had significantly lower rates of treatment failure at both 12 weeks and 24 weeks compared to the clinical observations (P = 0.0127, P < 1x10^-6, respectively). In contrast, the null and moderate strains had comparable predicted BV treatment failure rates at both time points. At 12 weeks, the clinically reported frequency was 34.6% versus the null strain (39.2%; P = 0.294) and moderate strain (36.2%; P = 0.703). At 24 weeks, the clinically reported frequency was 48.4% versus the null strain (40.7%; P = 0.0954) and moderate strain (42.4%; P = 0.1949). A negative control was evaluated to demonstrate that probiotics could perform worse than the null strain, which exhibited a 57.5% and 64.4% recurrence rate at 12 weeks and 24 weeks (Fig 4C). Altogether, these results suggest the Lactin-V probiotic strain (*L. Crispatus CTV-05*) has a competitive interaction with endogenous *Lactobacillus* spp. and the probiotic has a stronger impact on nAB ($\alpha_{P \to nAB}$) than nAB has on the probiotic ($\alpha_{nAB \to P}$) similar to our hypothesized moderate strain. Selection of probiotic strains with less competition with endogenous *Lactobacillus* spp. may help promote treatment efficacy

## Adjunctive antimicrobial therapy improves probiotic strain efficacy for underperforming strains

To systematically assess the importance of probiotic strain characteristics with respect to promoting *Lactobacillus* spp. dominant communities post-treatment, 500 *in silico* strains were evaluated. The 500 *in silico* strains were generated by Latin Hypercube Sampling of probability distributions similar to the reference virtual population. Each of the 500 strains was then evaluated on the 2,000 subject virtual patient population for both the short-term probiotic regimen (7 days, once daily dose of probiotic followed by no pre-treatment with antibiotic) and the Lactin-V regimen (antibiotic pre-treatment followed by 11-week probiotic regimen). Each strain was assigned to a designated response profile based on the strain's performance across the 2,000 subjects. A Partial Least Squares Discriminant Analysis was used to evaluate the relationship between model parameters and response profiles (Fig 5) [50]. For the short-term therapy without antibiotic pre-treatment, the most important parameters driving a *Lactobacillus* spp. dominant response were separated along latent variable 1 (LV1) with $\alpha_{nAB \to P}$ having the strongest association (Fig 5A). Separation across LV2 captured differences between *Lactobacillus* spp., with Li-dominant responses most associated with Li inhibiting probiotic (more negative $\alpha_{Li \to P}$) and probiotic promoting the growth of Li (more positive $\alpha_{P \to Li}$). In contrast, the Lactin-V regimen was more sensitive to the interaction of endogenous *Lactobacillus* spp., namely Li ($\alpha_{Li \to P}$, Fig 5B). Between the two regimens, there were several probiotic strains that were predicted to promote probiotic dominance across greater than 99% of the virtual patients.

Characteristics of strains that were best at promoting a certain response over the 2,000 virtual patient population were similar between the regimens (S4, S5 Figs). Strains most associated with treatment failure (nAB-dominant response profiles) typically had strong negative interactions of nAB on P ($\alpha_{nAB \to P}$). Strains that most frequently exhibited boosted endogenous *Lactobacillus* spp. were associated with probiotic having a positive impact on Li ($\alpha_{P \to Li}$) for Li response types and oLB ($\alpha_{P \to oLB}$) for oLB response types. Lastly, high probiotic abundance was associated with the probiotic inhibiting all resident vaginal microbiota ($\alpha_{P \to nAB}$, $\alpha_{P \to Li}$, $\alpha_{P \to oLB}$) and with the resident microbiota tending to have positive interactions ($\alpha_{nAB \to P}$, $\alpha_{Li \to P}$, $\alpha_{oLB \to P}$). To evaluate the impact of regimen on probiotic efficacy, 5 strains were evaluated with the Lactin-V regimen and the short-term probiotic regimen (Fig 6). The parameter

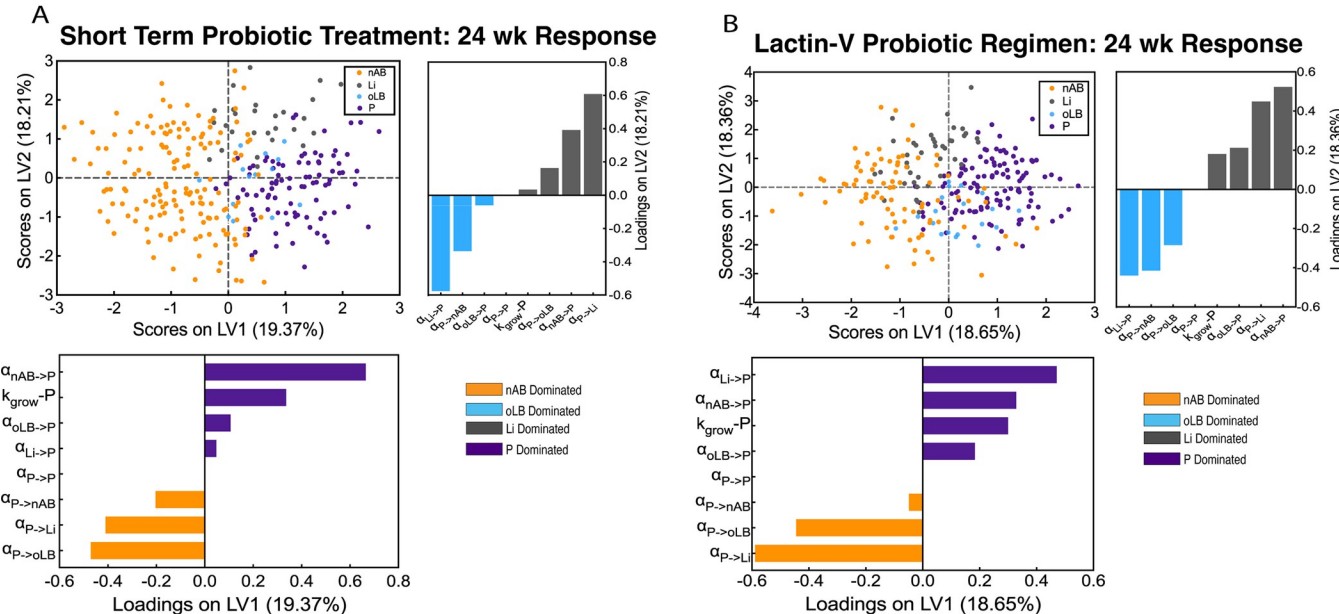

**Fig 5. Drivers of population level composition alterations for a short-term regimen and Lactin-V regimen.** Partial least squares discriminant analysis (PLS-DA) models were constructed using strain characteristics to predict a population-level profile type (consistently nAB, Li, oLB, or P-dominant across the 2,000 subject virtual population) after (A) short-term regimen of probiotic (7 days of once daily probiotic treatment) and (B) Lactin-V regimen of probiotic (5 days MTZ pretreatment followed by 11 weeks of probiotic treatment). In each regimen, the scores plots indicate how the five hundred hypothetical strains generated by Latin Hypercube Sampling (LHS) are distributed according to their response types in the new component space. The loadings plots show how much each parameter contributes to the separation of clusters across the corresponding latent variables, highlighting the influence of each parameter on the response types.

values for these 5 strains are summarized in S4 Table. The 5 strains selected were strains that promoted one of the four response types across the highest percent of the virtual population. For example, the strain for nAB response types elicited nAB-dominant in 92% of the virtual patients, the strain for Li response elicited Li-dominant in 86% of virtual patients, the oLB response strain elicited oLB-dominant in 74% of the virtual patients, and the probiotic response strain elicited a probiotic dominant response in 100% of the patients. The fifth strain was a strain that most closely replicated the Lactin-V phase 2b clinical trial results for BV recurrence at 12 and 24 weeks. Overall, the selected strains that promoted nAB-dominant responses, Li-dominant responses, oLB-dominant responses, and P-dominant responses had similar long-term effects on compositions for the Lactin-V regimen and the short-term regimen (Fig 6A–6B). This result was also observed when the top strains selected from the short-term regimen were tested with the Lactin-V regimen (S6 Fig). Effects differed for the representative strain for the Lactin-V formulation, where without the pre-treatment of antibiotic, the addition of the probiotic had little to no impact on re-orienting the vaginal microbiome composition. This result suggests that pre-treatment with antibiotics is critical for probiotics that cannot stably engraft into the existing vaginal community.

## Lactin-V efficacy is moderately dependent on probiotic dosing frequency and dose amount

Lastly, the Lactin-V probiotic dosing regimen was analyzed across 5 representative strains represented in S4 Table. These representative strains were identified by determining the 5 strains of the 500 LHS probiotic strains that had the most similar performance based on sum of squares error as *L. crispatus* CTV-05 when applied to the same phase 2b Lactin-V regimen and

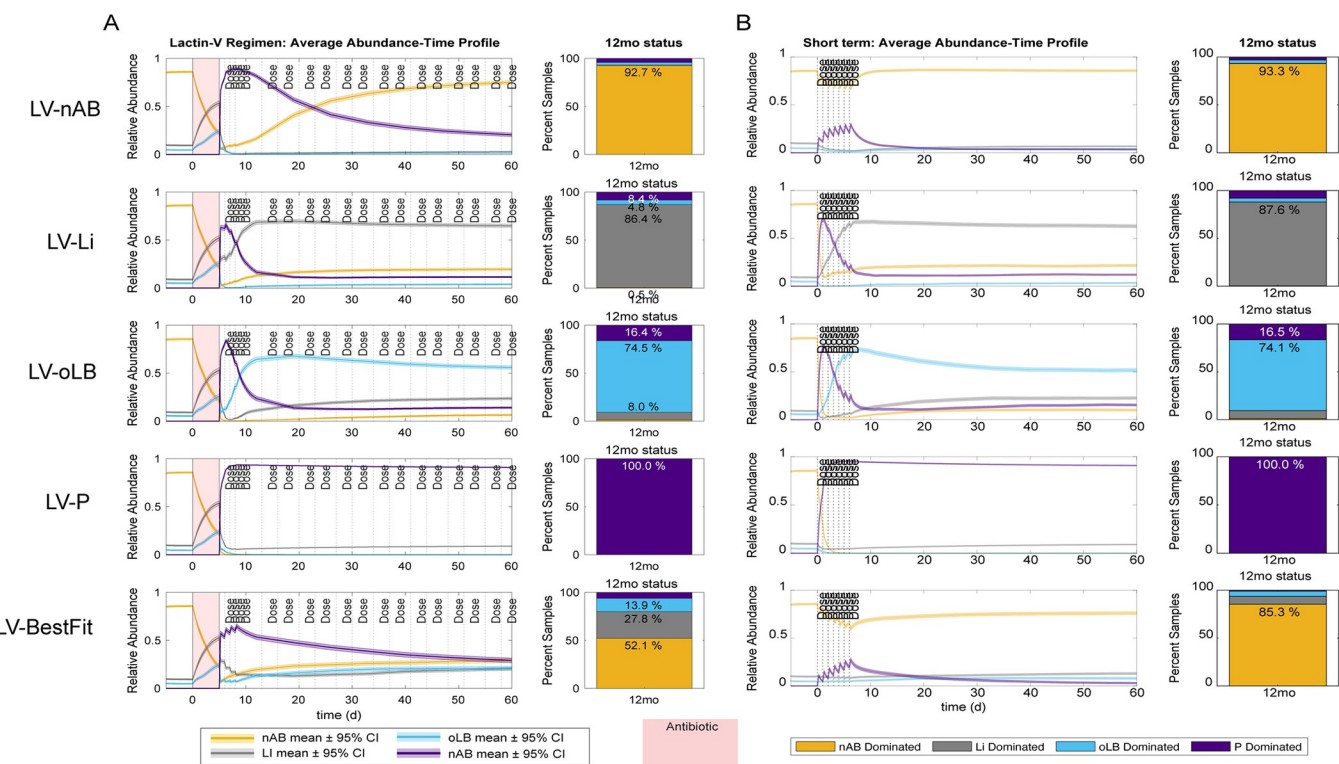

**Fig 6. Comparison of Lactin-V regimen with the short-term probiotic only regimen on virtual BV+ patients.** Parameter values are provided in S4 Table. (A) Lactin-V regimen abundance-time profile of community groups for each probiotic strain and predicted 12mo frequency of response types. (B) Short-term regimen abundance-time profile and predicted 12mo frequency of response types for strains defined in (A). Red indicates time of antibiotic dosing.

evaluated at 12 and 24 weeks. Using these strains, the dosing regimen during weeks 2–10 were altered to be, bi-weekly (1 dose every other week), once weekly, twice weekly (Lactin-V), four times weekly, and daily. There was a gradual decrease in treatment failure rate with increasing dose frequency (Fig 7A and 7B). Differences were more prominent at week 12, where the original Lactin-V regimen (twice weekly) had a statistically significant failure rate (40.0%) than the bi-weekly (47.3%) and daily (29.6%) dosing frequencies (P = 0.0165 and P = 0.0165, respectively; Fig 7A). At 24 weeks, results were more similar regardless of the dosing regimen and only the daily regimen differed from the less frequent dosing strategies (Fig 7B). Altering the dose amount, colony forming units (CFUs) per dose, was also evaluated at 12 weeks and 24 weeks for the phase 2b Lactin-V regimen (Fig 7C and 7D). All simulated changes to dose amount, starting from 5x greater for 5x less than the baseline dose resulted in significant changes in treatment failure rate (P < 1x10^{-6}, $\chi^2$ test). Altogether these results demonstrate that the dosing regimen for Lactin-V may be able to be further optimized for dose amount and frequency.

## Discussion

To examine characteristics of probiotic strains that can aid in re-orienting the vaginal microbiome from a nAB-dominant, BV-associated state, to a *Lactobacillus* spp. dominant state, we developed an *in silico* framework. This framework used generalized Lotka-Volterra ODEs to represent growth characteristics and interspecies interactions of three core community compositional types of the VMB. Probiotic strain characteristics were systematically evaluated across a virtual patient population, allowing for the investigation of probiotic parameters and

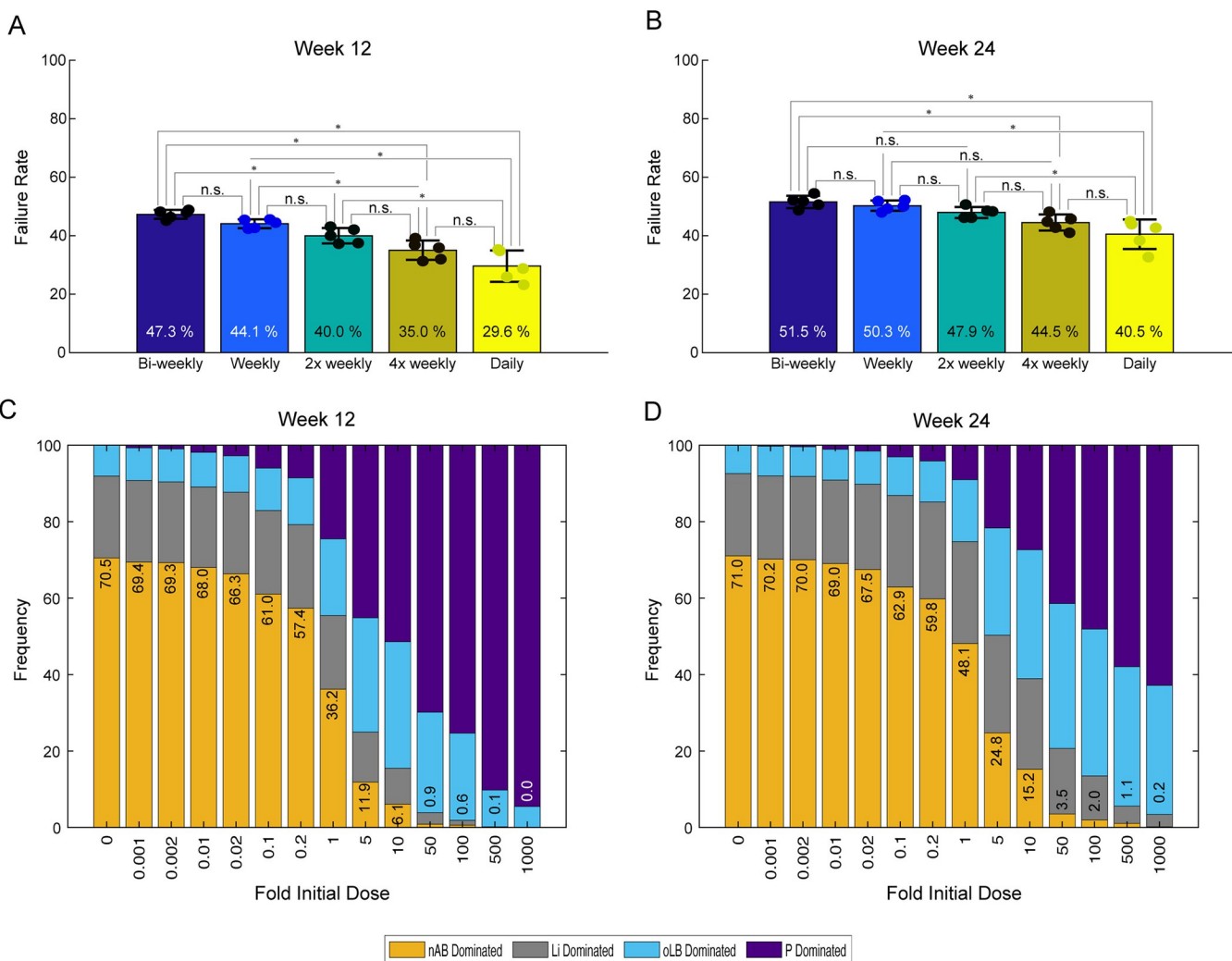

**Fig 7. Impact of alternative dosing frequency or dose amount on predicted BV recurrence rates in virtual BV+ patients.** The Lactin-V regimen has a 10 week period of intermittent dosing (2x weekly). Alternative dosing strategies for the 10 week period tested were bi-weekly, weekly, 4x weekly and daily. Treatment failure rate (recurrence rate) was evaluated at week 12 (A) and 24 (B). Dots indicate the recurrence rates of the 5 representative strains that recapitulated observed Lactin-V recurrence rates. Bars are the mean ± standard deviation. Multiple Mann-Whitney rank sum tests were used to compare groups and were adjusted using Benjamini & Hochberg procedure. (C-D) Impact of increasing or decreasing Lactin-V dose (CFUs) for the published Lactin-V regimen. The dose was modified 0.001x to 1000x the baseline dose and evaluated at week 12 (C) and week 24 (D).

probiotic regimens that elicit desired compositional changes that include Li-dominant, oLB-dominant, or probiotic-dominant communities.

Local and global sensitivity analyses implicated the importance of resident community members on probiotic efficacy. This result is not unsurprising, as it aligns with the ecological principle of community invasibility, which is driven by how the resident community members interact with the invading species (probiotic strain) [51,52]. Regimens without a course of antibiotics prior to probiotic were particularly sensitive to the impact of nAB on the probiotic strain ($\alpha_{nAB \to P}$). This parameter is interesting as few studies report the impact of nAB on *Lactobacillus* spp. or probiotic strains. One study that reports the pairwise impact of *G. vaginalis* with *L. jensenii* observed a 50% decrease in *L. jensenii* cell density from mono-culture to co-culture [53]. Another study reported that certain *G. vaginalis* strains could selectively inhibit the *L. crispatus* to adherence to epithelial cells over *L. iners* [54]. nAB associated with BV may

also inhibit the growth of *Lactobacillus* spp. through the production of biogenic amines, which were shown to increase *Lactobacillus* spp. lag-time, decrease growth rates, and decrease the production of D/L-lactic acid [41]. These results suggest that critical selection criteria for probiotic strains used to treat BV include the ability for the strain to outcompete a variety of nAB for adhesion to epithelial cells, disrupt biofilms, as well as be tolerant to biogenic amines.

For probiotic regimens that include pre-treatment with the first line BV antibiotic metronidazole, the impact of nAB on the probiotic strain becomes less important and the impact of endogenous *Lactobacillus* spp. is more prominent ($\alpha_{Li \to P}$, $\alpha_{oLB \to P}$). This result is because metronidazole only inhibits the nAB population, whereas *Lactobacillus* spp. are generally resistant [55]. Therefore pre-treatment with the antibiotic MTZ results in lower levels of nAB at the time of probiotic administration and a higher impact of endogenous *Lactobacillus* spp. that are present when probiotic is administered, which is more commonly *Li* in BV+ individuals and in this virtual patient population. Notably, colonization with endogenous *Lactobacillus* spp. is reported to decrease probiotic (*L. crispatus* CTV-05) in healthy individuals [56], supporting the importance in considering probiotic strain interactions with *Lactobacillus* spp. Despite the importance of interactions between resident *Lactobacillus* spp. and the probiotic strain, little is known about the pairwise relationship. A publication analyzing 17 *L. crispatus* strains, two *L. gasseri* strains, two *L. jensenii* strains and a *L. iners* strain using spot agar tests demonstrated that all human strains of *L. crispatus* could slightly inhibit *L. iners* growth [57]. Not all strains of *L. gasseri* and *L. jensenii* inhibited *L. iners*. Co-culture data demonstrated that some *Lactobacillus* strains had a 4–6 log reduction in cells with certain *L. crispatus* strains [57]. Thus, screening of the relationship between probiotic strains and endogenous *Lactobacillus* spp. could help improve probiotic design.

Boosting endogenous levels of optimal species like *L. crispatus*, *L. gasseri*, or *L. jensenii* (oLB) could be preferential to stably incorporating a probiotic strain, particularly if the strain is not native or known to have the same functional capacity in the vaginal microbiome as endogenous *Lactobacillus* spp. This framework indicated for endogenous oLB to become dominant after treatment, probiotic strains must have a facilitative interaction with oLB (positive $\alpha_{P \to oLB}$), inhibitory interaction with Li (negative $\alpha_{P \to Li}$), and oLB must have a weak positive or inhibitory (negative $\alpha_{oLB \to P}$) impact on the probiotic strain. A potential oLB-promoting probiotic strain could be one that selectively inhibits L. iners, such as the human intestinal Lactobacillus spp. strain (L. paragasseri K7) and a less potent vaginal strain (L. gasseri 105–1) [49]. Combinatorial therapy that selectively targets L. iners could potentially enhance a probiotic strain that may not inherently have selectivity against L. iners. Recent studies have suggested that targeting the cysteine dependence of L. iners could reduce competition between L. iners and oLB [48]. Overall, these findings further support the idea that increased characterization of the interactions between probiotic strains and endogenous Lactobacillus spp. could significantly improve probiotic design. Probiotic strategies can thus either directly inhibit L. iners or indirectly reduce its competitiveness through specific metabolic dependencies.

This framework replicated the results of the phase 2b clinical trial for Lactin-V and indicated that *L. crispatus* CTV-05 likely exhibits competitive interactions with resident community members. Of *in silico* strains that could replicate the Lactin-V trial results at 12 and 24 weeks, most were predicted to not stably integrate into the community. Additionally, the efficacy of these strains was highly dependent on pre-treatment antibiotic, supporting the need for adjunctive use of antibiotics with Lactin-V. The model is more accurate at predicting the long-term impact of antibiotic therapy (24 weeks) than short-term (12 weeks), likely because it is less accurate at capturing the initial transient impact. In published randomized control trials, dosing frequency is variable and often not clearly justified (S2 Table). To evaluate the impact of dose frequency for Lactin-V, dosing was simulated bi-weekly, once weekly, twice weekly

(Lactin-V), four times weekly, and daily and the frequency of BV recurrence was evaluated at 12 and 24 weeks. Dosing frequency had a moderate impact on probiotic efficacy evaluated at 12 weeks, but less of an impact at 24 weeks, which is 3 months after subjects finished therapy. The importance of dosing frequency is likely strain dependent, with strains that are unable to engraft into the community most dependent on frequent dosing. Likewise, our simulations demonstrate a dose-dependent relationship with treatment success rates. This framework could help guide, optimize, and provide rationale for future probiotic regimen designs.

A major limitation of this model is that it does not account for variability in host behavior. Factors such as hormonal fluctuations due to menstrual cycles, birth control, or pregnancy can impact the stability and composition of the vaginal microbiome over time [13,43,58–60]. Sexual and hygienic behavior would also be predicted to impact composition by introducing or removing species present and changes in vaginal pH [56,61,62]. Additionally, this methodology is a reductionist approach to recapitulating the vaginal microbiome and capturing species interactions. Species interactions are likely dependent on the competition for limited substrates as well as cross-feeding, which could be incorporated into the model using methodologies like Monod equations [63–65]. Future iterations of the model that incorporate these details could be personalized to subject behavior and metabolic microenvironment.

Overall, this work provides a new framework to characterize and predict how probiotic strain characteristics contribute to compositional changes during and after treatment cessation. An *in silico* framework to test probiotic strains is particularly important for the vaginal microbiome, as standard *in vitro* and *in vivo* models fail to replicate the base characteristics of the vaginal microenvironment such as co-existence of appropriate vaginal microbiota or low vaginal pH [66,67]. Moreover, current therapeutic regimens to modulate vaginal microbiome composition have high rates of treatment failure, emphasizing the need to develop better tools to evaluate alternative therapies [68]. This framework could be particularly informative in combination with newly developed vagina-on-a-chip technologies to effectively screen new probiotic strains [66]. Together, the use of *in silico* models and new developments in experimental technologies will inform the rational selection of probiotic strains and intelligent design of dosing regimens with or without adjunct antimicrobial use.

## Materials and methods

### Model construction

A generalized Lotka-Volterra model (gLVM) with three equations was used as the ordinary differential equation-based model. gLVMs include the growth rate of each species ($k_{grow}$), the self-interaction term (contributes to carrying capacity: $\alpha_{x \to x}$) and inter-species interaction terms ($\alpha_{x \to y}$). Growth rates were always assumed to be positive when the system is not under any perturbation, such as menses or antibiotic therapy. self-interaction terms were assumed to always be negative, and the inter-species interaction terms could be either positive or negative. For the addition of the probiotic into the model, a fourth microbial population was created resulting in the following equations. All model simulations were completed in MATLAB 2020b and are published at: https://github.com/chyylee/CST_Probiotic

$$\frac{d[nAB]}{dt} = k_{grow-nAB}[nAB] + \alpha_{nAB \to nAB}[nAB][nAB] + \alpha_{Li \to nAB}[nAB][Li] + \alpha_{oLB \to nAB}[nAB][oLB] + \alpha_{P \to nAB}[nAB][P]$$

$$\frac{d[Li]}{dt} = k_{grow-Li}[Li] + \alpha_{Li \to Li}[Li][Li] + \alpha_{nAB \to Li}[Li][nAB] + \alpha_{oLB \to Li}[Li][oLB] + \alpha_{P \to Li}[Li][P]$$

$$\frac{d[oLB]}{dt} = k_{grow-oLB}[oLB] + \alpha_{oLB \to oLB}[oLB][oLB] + \alpha_{nAB \to oLB}[oLB][nAB] + \alpha_{Li \to oLB}[oLB][Li]$$
$$+ \alpha_{P \to oLB}[oLB][P]$$

$$\frac{d[P]}{dt} = k_{grow-P}[P] + \alpha_{P \to P}[P][P] + \alpha_{nAB \to P}[P][nAB] + \alpha_{Li \to P}[P][Li] + \alpha_{oLB \to P}[P][oLB]$$

For simulations involving antibiotic treatment with metronidazole, the net growth rate was set to be negative (kkill) resulting in a first order decay rate based on values reported in Mayer et al. 2015 [69]. This parameter change was set to impact nAB group for the period associated with duration of the antibiotic regimen.

$$\frac{d[nAB]}{dt} = kkill[nAB] + \alpha_{nAB \to nAB}[nAB][nAB] + \alpha_{Li \to nAB}[nAB][Li] + \alpha_{oLB \to nAB}[nAB][oLB] + \alpha_{P \to nAB}[nAB][P]$$

## Virtual population development

To test the impact of a probiotic strain at the population level, a virtual patient population was generated using Latin Hypercube Sampling of physiologically defined parameter ranges as described in Lee *et al.* 2023 [42] (S1 Table). Briefly, a large collection of virtual patients was generated (30,000) and then resampled to match clinically observed CST equilibrium behavior distribution pattern of the Human Microbiome Project Cohort (HMP) described in Lee *et al.* 2023. The CST equilibrium behavior describes the stability in CST classification over time, where subjects that consistently exhibit the same CST are considered mono-stable (1SS) and those that switch between different CSTs are considered multi-stable (2SS). Model predictions were related to these CST equilibrium behaviors using a nearest centroid classifier of the predicted relative abundances, and whether the system exhibited mono or multiple equilibrium states referred to as CST equilibrium behavior. The centroids for the nearest centroid classifier were determined from VALENCIA as described in Lee *et al.* 2023 [14,42]. Probiotic strains were tested on the subset of 2,000 HMP cohort virtual patients that could obtain a nAB-dominant (BV positive) state at equilibrium which includes the 1SS nAB-dominant (60%), 2SS nAB-dominant / Li-dominant (31%), and 2SS nAB-dominant / oLB-dominant CST (9%) equilibrium behaviors. The HMP cohort had a similar frequency of self-identified White/Caucasian subjects relative to persons of color as the Lactin-V cohort (32% versus 35%, respectively) [43].

## BV treatment regimens

The probiotic dose was calibrated to the relative abundance distribution observed in Dausset et al. [30], 1 day after an initial dose was given and kept constant throughout the manuscript. Two main probiotic regimens were evaluated: a short-term probiotic therapy without antibiotic pre-treatment and a long-term probiotic therapy with antibiotic treatment modeled after the regimen for a phase 2b study of Lactin-V [29]. The short-term therapy included a 7-day, once daily dosing of probiotic strains. The Lactin-V regimen included a 5-day antibiotic regimen followed by 4-days of daily dosed probiotics and then 10-weeks of twice weekly probiotic doses. Evaluation of the impact of probiotic was evaluated at multiple time points. The 5-day antibiotic regimen was simulated as a negative impact on nAB growth rate at magnitudes calculated from Mayer et al. [69]. For the short-term probiotic, impact on composition was evaluated at treatment cessation, 1 week, 1 month, 3 months, 6 months, and 12 months. For the

Lactin-V regimen, impact on composition was evaluated at 12 weeks (1 week after therapy cessation) and 24 weeks (3 months after therapy cessation). Impact of composition was assessed by classifying the CST after treatment using a nearest centroid classifier. Possible classes were nAB-dominant (BV+), Li-dominant, oLB-dominant, or P-dominant. Regimens were simulated across 2,000 patients, and the frequency of each response type was reported per strain tested.

## Local sensitivity analysis

Local sensitivity analyses were centered at the parameters for a null probiotic strain and evaluated at 12 months post treatment cessation. The null probiotic strain was defined as a probiotic strain that had a moderate growth rate ($0.5$ $d^{-1}$) and negligible interspecies interactions ($\alpha_{nAB\to P}$, $\alpha_{Li\to P}$, $\alpha_{oLB\to P}$, $\alpha_{P\to nAB}$, $\alpha_{P\to Li}$, $\alpha_{P\to oLB}$ = $0.0$ density$^{-1}$d$^{-1}$) and a moderate self-interaction term ($\alpha_{P\to P}$ = $-0.022$ density$^{-1}$d$^{-1}$). For the 1-dimensional (1D) perturbation analysis, each probiotic strain parameter was altered one-at-a-time over a set parameter range. The range was $-0.10$ to $0.10$ density$^{-1}$d$^{-1}$ for the interspecies interaction terms, $0$ to $1.0$ d$^{-1}$ for the growth rate, and $-0.004$ to $-0.04$ density$^{-1}$d$^{-1}$ for the self-interaction term. Sensitivity was quantified using a normalized sensitivity metric [44]. This metric is $S_{ij} = (\Delta Y_i / \Delta P_j)(P_{o,j}/Y_{o,i})$, where i is the ith parameter and j is the jth response metric. Y indicates the response value for the parameter change, $Y_0$ refers to the value with no change to the parameter (null strain). $P_0$ is the initial value of the parameter. $Y_0$ is the outcome based on $P_0$. For inter-species parameters we used $P_0$ = $0.01$, and adjusted $Y_0$ accordingly. For the four-parameter perturbation analysis, up to four parameters were modified at a time. Two parameters that were most sensitive to altering probiotic efficacy (changes in rate of nAB-dominant states) and two parameters most sensitive to altering frequency of Li/oLB-dominant states were selected. Parameters could undergo a positive change (+$0.01$ density$^{-1}$d$^{-1}$), no change ($0.00$ density$^{-1}$d$^{-1}$), or negative ($-0.01$ density$^{-1}$d$^{-1}$) alteration for each parameter, in combination (3 values$^{4\ parameters}$ = 81 combinations).

## Systematic probiotic strain selection

A global uncertainty analysis was used to systematically evaluate probiotic strain characteristics that could consistently promote nAB-dominant, Li-dominant, oLB-dominant, or P-dominant communities across the 2,000 subject virtual population. Latin Hypercube Sampling was used to generate 500 *in silico* candidate probiotic strains. Parameter values were sampled from uniform distributions defined with the same ranges used to create the virtual population, excluding $\alpha_{P\to nAB}$, which was constrained to be negative. Each of the 500 candidate strains was tested in the framework for the short-term regimen and the Lactin-V regimen. Each strain was then classified by the response type that occurred in the highest frequency across the virtual population. For example, by testing a probiotic strain on 2000 virtual samples, if 10% of patients become nAB-dominant, 20% Li-dominant, 60% oLB-dominant, and 10% P-dominant, the strain would be classified as a oLB-promoting strain. The association between probiotic strain characteristics and the response classification was assessed using Partial Least Squares Discriminant Analysis [50]. The top 5 strains that promoted nAB, Li, oLB, and P-dominant states for each regimen were visualized and compared against the other regimen. To understand the Lactin-V regimen, the top 5 *in silico* strains that had similar BV recurrence rates at 12 and 24 weeks were identified by the sum of absolute distance from the predicted and observed rates.

## Statistical analyses

All statistical analyses were completed in MATLAB. Chi-square tests were used to compare frequencies of response types between groups. Wilcoxon rank sum tests were used to compare

numerical data. Where noted, P-values were adjusted using Benjamini & Hochberg procedure [70]. The PLS-DA model was created using the PLS Toolbox in MATLAB 2017b using 10-fold cross-validation. Briefly, PLS models are a supervised approach that assigns a loading to each feature (probiotic strain characteristics) and identifies a linear combination of loadings that best separates the response variable (nAB, Li, oLB, P-promoting classification). The linear combination of loadings is referred to as a latent variable (LV) and indicates the magnitude of association between a feature and the response group.

## Supporting information

**S1 Table. Explanation of LHS parameter ranges.** Note the determination of inter-species interaction terms was based on empirical observation and hypothesis on interaction term strength and directionality. More information can be found in a previous publication in Supplementary Note 1 [42]. Values were scaled to *in vivo* rates based on Stein et al. [37]. This table also includes references [71–75].
(DOCX)

**S2 Table. Clinical Probiotic Regimens.** This table details various probiotic dosing regimens, including whether pretreatment with antibiotics was administered, the routes of administration (vaginal, oral, or both), the length of each regimen, the specific probiotic strains used, the final results, the bacterial vaginosis (BV) evaluation metrics, and the final end points. The diversity in these probiotic regimens demonstrates the variability in outcomes observed across different clinical trials. This table also includes references [76–94].
(DOCX)

**S3 Table. Local Sensitivity Analysis of Probiotic Strain Characteristics and Their Impact on Each Response Type.** The sensitivity of each parameter and response type (nAB, Li, oLB, P-dominant) was measured using a method from previous research, which standardizes the changes in outcomes relative to the variations in parameter values [44].
(DOCX)

**S4 Table. Parameter values of the hypothetical strains used in Figs 6 and 7.** The Latin Hypercube Sampling (LHS) method was used to generate 500 in silico strains. Among these, the top strains (nAB, Li, oLB, P) were identified as those that elicited one of the four response types in the highest percentage of the virtual population. The Best Fit strains, designated as Latin-V, V2, V3, V4, and V5, were the top 5 strains that most closely matched the performance of L. crispatus CTV-05. This comparison was based on the sum of squares error and was carried out using the same phase 2b Lactin-V regimen, with evaluations conducted at 12 and 24 weeks.
(DOCX)

**S1 Fig. Overview of analysis methodology.** A virtual population was generated based on the Human Microbiome Project cohort. The analysis included two regimen types, a short-term probiotic regimen with no antibiotic pre-treatment and the regimen described in a phase 2b Lactin-V clinical trial (Cohen et al. 2020). In this framework, we tested various hypothetical probiotic strains, focusing on their interaction parameters with endogenous species. Additionally, we simulated our model using different probiotic and antibiotic regimens. Through these simulations, we identified the most important probiotic design criteria to address the issue of recurrent BV.
(TIF)

**S2 Fig. Four parameter local sensitivity analysis.** The top sensitive parameters for BV clearance and top parameters for modulating endogenous *Lactobacillus* spp. levels were modified systematically from -0.01, 0.00, +0.01 density$^{-1}$d$^{-1}$ from the null probiotic strain parameter values giving rise to 81 possible parameter combinations (top heatmap). The percent of the 2,000 *in silico* BV+ subjects that elicited a certain response type at 12months post therapy cessation were visualized (nAB-dominant, Li-dominant, oLB-dominant, or P-dominant). Data is plotted from most efficacious (left) to least (right). The null probiotic strain is indicated by "B>" and the red line. A chi-square test was performed to identify whether there is a significant difference in terms of efficacy between each of the 81 possible parameter combinations and the null strain. Asterisks indicate a significant change in efficacy relative to the null probiotic strain. (TIF)

**S3 Fig. Lactin-V phase 2b clinical trial versus model simulations with increased antibiotic effect.** Comparison of model predictions with Lactin-V trial results at 12 and 24 weeks for the placebo and the treatment arm. For the treatment arm, 4 strains were simulated by the model encompassing a traditionally designed probiotic, null probiotic, moderately/conservatively designed probiotic, and bad/negative control probiotic. The impact of antibiotic was simulated at a magnitude (-3.82 d^-1) that was equivalent to the most sensitive *G. vaginalis* strain in Mayer et al. 2015 [69].
(TIF)

**S4 Fig. Parameter values for strains that could elicit consistent population-level compositional changes for a short-term regimen.** Strains in the 90[th] percentile or higher for imparting a certain compositional effect across the virtual population are visualized. (A) Consistently promote nAB-dominant communities (B) Li-dominant communities (C) oLB-dominant communities (D) P-dominant communities.
(TIF)

**S5 Fig. Parameter values for strains that could elicit consistent population-level compositional changes with the Lactin-V regimen.** Strains in the 90[th] percentile or higher for imparting a certain compositional effect across the virtual population are visualized. (A) Consistently promote nAB-dominant communities (B) Li-dominant communities (C) oLB-dominant communities (D) P-dominant communities.
(TIF)

**S6 Fig. Comparison short-term probiotic only regimen top strains with Lactin-V regimen.** (A) Interspecies interaction parameter values of top probiotic strains for the short-term probiotic only regimen. (B) Abundance-time profile of community groups for each probiotic strain and predicted 12mo frequency of response types. (C) Lactin-V regimen abundance-time profile and predicted 12mo frequency of response types for strains defined in (A). Red indicates time of antibiotic dosing.
(TIF)

## Author Contributions

**Conceptualization:** Christina Y. Lee, Kelly B. Arnold.

**Formal analysis:** Christina Y. Lee, Sina Bonakdar.

**Funding acquisition:** Kelly B. Arnold.

**Methodology:** Christina Y. Lee, Kelly B. Arnold.

**Project administration:** Kelly B. Arnold.

**Supervision:** Kelly B. Arnold.

**Validation:** Christina Y. Lee, Sina Bonakdar.

**Visualization:** Christina Y. Lee, Sina Bonakdar.

**Writing – original draft:** Christina Y. Lee, Kelly B. Arnold.

**Writing – review & editing:** Christina Y. Lee, Sina Bonakdar, Kelly B. Arnold.

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
