## [Decision Letter · Decision Letter 0]

8 Aug 2024

Dear Dr. Arnold,

Thank you very much for submitting your manuscript "An in silico framework for the rational design of vaginal probiotic therapy" for consideration at PLOS Computational Biology.

As with all papers reviewed by the journal, your manuscript was reviewed by members of the editorial board and by several independent reviewers. In light of the reviews (below this email), we would like to invite the resubmission of a significantly-revised version that takes into account the reviewers' comments.

We cannot make any decision about publication until we have seen the revised manuscript and your response to the reviewers' comments. Your revised manuscript is also likely to be sent to reviewers for further evaluation.

Sincerely,

Roger Dimitri Kouyos

Academic Editor

PLOS Computational Biology

Amber Smith

Section Editor

PLOS Computational Biology

Reviewer's Responses to Questions

**Comments to the Authors:**

Reviewer #1: The authors developed a comprehensive in silico framework to guide the rational design of more effective vaginal probiotic therapies. By simulating the complex microbial interactions in the vaginal ecosystem, the presented model can identify key parameters that govern the stability and resilience of different vaginal microbiome states.

Positive aspects:

- It provides a quantitative approach to understand vaginal microbiome dynamics

- It can help identify critical microbial interactions and parameters that influence microbiome stability

- It has the potential to guide the development of more targeted and effective probiotic interventions

Potential limitation/negative aspect:

- The model relies on simplifications and assumptions about the vaginal microbiome that may not fully capture its complexity

Main comments

My main concern centers on the authors' treatment of species proportions during evaluation. The focus seems solely on the dominant specie, regardless of its proportion (e.g. 99% or 51%). This overlooks the case where the second most abundant species might be nearly as prevalent as the dominant one. Such close proportions between species don't seem to be considered in the analysis. This approach lacks clear justification in the paper, and the potential implications of this choice are left unexplored. I believe there's a need to refine this measurement methodology to better capture the species distribution and their potential impact. Having a diversity measure, such as an entropy measurement, could clarify the distribution at each time point.

Regarding the success associated with Li, oLB, or Probiotic dominance, the introduction clearly distinguishes Li as less beneficial to health. Given its differences, I don't think it's appropriate to consider Li dominant and oLB dominant as equally successful outcomes. The rationale behind this equivalence isn't adequately explained in the paper, I suggest mentioning this point at least in the discussion.

Additionally, there are instances where the reasoning for selecting a specific evaluation time point for particular regime is unclear. For example, in Figure 5, two regimes are presented at different end states without clear justification for the choice of these specific time points. For a meaningful comparison, these regimes should be depicted at the same end state to ensure a fair evaluation.

In general, I think the paper would greatly benefit from more intuitions for the model parameters, i.e., what the inter-species interaction terms represent. This would give a clearer model understanding.

Further detailed comments and suggestions are included in the attached PDF.

Reviewer #2: Here the authors used model prediction to assess probiotic therapies from a phase 2b trial with Lactin V in the treatment of bacterial vaginosis. Overall the authors found that antibiotic therapy and increasing dosage of probiotics may prevent BV recurrence. This is a comprehensive paper and incredibly important for this highly understudied field of women’s health. Minor comments below.

1. The abstract could be more clear that this is a modeling/simulation driven paper and what is being used.

2. The first paragraph of the results is unclear how the simulation actually is representative of “real life” scenarios, please clarify of possibly make a cartoon figure to make this more clear to the reader. Figure 1B attempts this but the “virtual patient” concept is confusing to this biologist.

3. In figure 3, the stats *** make the figure very busy. Perhaps make a small table of statistical comparisons instead.

4. The figure legends should indicate if the data is based on virtual vs real patients

Reviewer #3: Authors develop a computational framework to assess the efficacy of vaginal probiotic treatment regimens to combat Bacterial vaginosis. They develop a mechanistic model based on gLV dynamics and incorporate the impact of antibiotic use in combination with the impact of species competition to understand the community composition after the application of various treatment regimens. I appreciate the work the authors put it, and I fundamentally believe in their approach in supporting research in probiotic design. However, the manuscript in its current form is confusing to the reader. One of my major concerns is the ambiguity around methods, especially about the choice of parameter distributions and how the model is calibrated to recapitulate the results of the clinical trials and the antibiotic treatment. Seems like the parameter ranges are derived from the previous publications in the literature (Table S1), but the steady state community composition is "related" to the clinical data using a nearest centroid classifier? Another major concern is the lack of quantitative support for the interpretation of the results. Authors use adjectives like "high/low" degree of sensitivity without proper methods to support these comparisons. There is also a general lack of citations/references in the main text and figures captions are not sufficient. Please find my detailed comments below.

Abstract

"Bacterial vaginosis is a common condition characterized by a … ". Use the abbreviation BV here, as it appears in the rest of the abstract.

Introduction

References for the community state types? CSTs, CST -I, CST -II, CST -V are not very clear to the reader who is not very familiar with the VMB.

L. iners is "less" associated with health and commonly associated with BV? This sentence should be more clear, and needs a reference.

The "Non-optimal" part of the nAB should be briefly defined in the introduction. Same for "optimal". What do these terms mean exactly? Pathobiont vs symbiotic commensal?

Results

Reference for the method "Latin Hypercube Sampling"? In general, I appreciate the references in the supplementary but the main text needs better references. I will not point them out after this point, please go through the main text again and provide references for scientific/clinical statements and methods.

Right after the first paragraph of results, I still don't know what "probiotics" contain. Is it oLB? Is it any other Lactobacillus spp.? It makes it hard to interpret the CST classifications you defined to quantify the efficacy of the treatment.

What is a null and what is a traditional probiotic strain? What do these groups contain in terms of species?

How do you decide whether the growth rate of 0.5d-1 is moderate? Or self-interaction term of -0.022 density-1d-1? What is the rationale behind these quantitative cutoffs?

P-dominated or P-dominant? Pick one and stick with it for the entire text.

'The traditional strain had a lower rate of treatment failure, with nAB dominant communities comprising 32% of virtual patients for times greater than 1 month post-treatment cessation (P < 1x10-6, χ2 test between null and traditional at each timepoint 1 month and greater). ' -> this sentence is a bit confusing. When are you sampling the microbiome composition post-treatment? I assume the composition is dynamic, moving toward a new equilibrium after treatment cessation. This means depending on the time of sampling, you will see different percentages of nAB within the community composition. Similarly, with the chi-square test what are you comparing exactly? I assume it is the binary indicator of failure vs success since you are using chi-square?

In general, I am a bit confused about the null strain. You put a species that is not interacting with any other species, but it is still growing. So it has a niche that is not occupied by all the rest of the microbiome? I understand that you are trying to create an analogy to the "null" hypothesis here, but what you are trying to show is anyway about the "degree" of P-dominancy given how much P can inhibit the growth of nAB, no? If so, why don't you play with that parameter on a continuous scale and plot 12mo P dominated percentage vs alpha_P->nAB? This would be much better to understand how much inhibition one needs based on the target community composition at 12 mo. (As I continue reading, I started to understand that P compartment is "hypothetical", and does not contain any particular species yet, am I correct? This should really be clear to the reader at the beginning).

As I move, I see now my comment above is addressed in the parameter sensitivity analysis. I still keep it because this is how the reader thinks before they move on with the manuscript :) Given that the manuscript focuses on the interaction terms so much in the abstract and the introduction, I think you don't really need the null strain. You have all the information in the sensitivity analysis anyway.

You mention the inhibitory compounds (D/L-lactic acid, hydrogen peroxide, or bacteriocins) during the results section, but in the introduction there is not much about the efficacy of these products, rather you only mention that L. iners is not capable of producing them. There is really no literature around the use of probiotics that produce these metabolites?

Regarding the interpretation of the local sensitivity analysis:

The "degree" of sensitivity is not quantified here, therefore you cannot use phrases like "more/most sensitive parameters" or "high degree of sensitivity" in the text. You need to quantify this first. You can employ something like random forests and predictor importance analysis given you have a high dimensional model and options for both a binary outcome of treatment failure/success or the degree of nAB domination.

"The importance of probiotic inhibition of nAB (nABP) is unsurprising, given the selection of probiotics that produce inhibitory compounds (D/L-lactic acid, hydrogen peroxide, or bacteriocins) for nAB is a standard practice [26–30]. In contrast, the high degree of sensitivity for nAB on probiotic (nABP) is less intuitive and not well characterized in vitro or in vivo." -> there is a typo here. You use the same interaction term for probiotic inhibition of nAB and nAB on probiotic. This is super confusing.

Figure 2 needs a better figure caption. When is the time of sampling here? 12mo? (same goes for Fig. S2).

Figure S2: All treatments are significantly better than the null strain with three stars? How is the efficacy compared here? This is the supplementary, you can afford to explain more in the figure caption and make these figures standalone.

What is a "select" strain? Are these "real" species or are these hypothetical parameter combinations? It's very confusing!

"All selected strains were significantly from the null strain."? Significantly what?

Define MTZ again in the results before using the abbreviation.

Where does Lactin-V probiotic fall within all these ranges of hypothetical probiotics you have simulated so far in the manuscript? Since the manuscript focuses a lot on the interaction parameters' influence on the outcomes, the section "Combinatorial regimens can lower BV recurrence rates" should provide some quantitative idea of what those parameters are for the Lactin-V probiotic treatment.

Typo: "The simulation of the Lactin-V treatment arm included *for* possible probiotic strains"

I am really lost about the simulations for the Lactin-V regimen. Isn't it one strain, Lactobacillus crispatus CTV-05? Why do you simulate multiple strains?

Sorry if I missed this, but the quantitative impact of ABX (the killing rate) on nAB, Li, oLB, and P is stated exactly where in the manuscript? It is hard to interpret the results of Figure 4 without that information.

Figure caption of Fig. 5 is not sufficient. Also, what are "scores" and "loadings" on the y axis? Loadings mean eigenvalues? I see that this is in the statistical analysis, but the reader shouldn't have to go there to read Fig. 5. In general, please make all the figures standalone. The caption must be sufficient to understand and interpret the figure.

Fig. 6 is hard to read in general. One really needs to zoom in.

Discussion

Could you discuss why you have a statistically significant difference between the clinical results and your model in Fig. 4B at 12 weeks, despite having non-significant differences at 24 weeks? Whatever is different, it is transient in your model, which is a bit weird given that you give placebo. Since there is no perturbation to the system, it means there is something about the interaction/growth terms that lead to a similar steady state (24 weeks) but a transient difference in 12 weeks? If it were to be because of the interaction terms, then the steady state response should also be significantly different, right? Given that it is not, I wonder what creates this transient difference during the placebo.

It would be great to hear more discussion about mechanistically "why" the pre-treatment + probiotics work better than just probiotics. It's common practice to wipe out the microbiome first including the pathogen and then properly build it up. Authors start this paragraph "For probiotic regimens that include pre-treatment with antimicrobials like metronidazole of clindamycin, the impact of nAB on the probiotic strain becomes less important and the impact of endogenous Lactobacillus spp. is more prominent" but then this is actually not followed up properly. Antibiotics kill Lactobacillus spp. , give space to L. crispatus CTV-05 to do its job, and then Lactobacillus spp. grow back again? If so, does that mean we need to create a transient dysbiosis in the vaginal microbiome to allow L. crispatus CTV-05 dominance? What does this mean for colonization resistance against other infections in general?

"This framework indicated for endogenous oLB to become dominant after treatment that probiotic strains must have a facilitative interaction with oLB (positive P->oLB), inhibitory interaction with Li (negative P->Li), and oLB must have a weak or inhibitory (negative oLB->P) interaction with the probiotic strain." -> "weak or inhibitory"? Is it weak "and" inhibitory? Also, I don't really understand how authors came to this conclusion quantitatively. Where in the results is this indicated? Say both P->oLB and oLB->P are negative, but P->oLB is smaller in magnitude. Wouldn't it still work? I am very sorry if I missed that plot, but couldn't see it.

I think the end of the 4th paragraph of the discussion should be better organized in terms of ideas. Are you trying to say that you can have probiotics either directly inhibiting L. iners or indirectly inhibiting it by supporting its other endogenous competitors?

Methods

"For example, if a strain was 10% nAB-dominant, 20% Li-dominant, 60% oLB- dominant, and 10% P-dominant the strain would be classified as a oLB-promoting strain." -> It's not the strain that is dominant right, it's the distribution of results after 12mo (or whatever the time frame is) using this strain as the probiotics? This sentence should be more clear.

Regarding the mathematical model: where are the death rates when antibiotics are applied? Do you reduce the growth rate?

"These seven states were related to clinical data using a nearest centroid classifier of the predicted relative abundances." What does this mean? Did you fit your data or determined the parameter distributions given these steady state values? If so, what is the fitting procedure? If not, does this mean that given the distributions you used, you observed that they are "related"? Or picked the related ones among your virtual population? Because later you say "The virtual patient population was selected to match the CST equilibrium behavior distribution pattern of the Human Microbiome Project Cohort (HMP) described in Lee et al. 2023." I am really confused about the methodology here.

Supplementary

Figure S1 is a bit confusing, especially when the reader goes there right after the first paragraph of the results. This is in the supplementary anyway so you can afford to go a bit longer in the figure caption and make it easier to interpret.

In general, all figures in the main text and the supplementary should be standalone.

**Have the authors made all data and (if applicable) computational code underlying the findings in their manuscript fully available?**

Reviewer #1: Yes

Reviewer #2: Yes

Reviewer #3: Yes

PLOS authors have the option to publish the peer review history of their article (what does this mean?). If published, this will include your full peer review and any attached files.

Reviewer #1: No

Reviewer #2: No

Reviewer #3: No
---

## [Decision Letter · Decision Letter 1]

16 Jan 2025

Dear Dr. Arnold,

We are pleased to inform you that your manuscript 'An in silico framework for the rational design of vaginal probiotic therapy' has been provisionally accepted for publication in PLOS Computational Biology.

Best regards,

Amber Smith

Section Editor

PLOS Computational Biology

Reviewer's Responses to Questions

**Comments to the Authors:**

Reviewer #1: Thank you for your revisions and for addressing the reviewers' feedback.

**Have the authors made all data and (if applicable) computational code underlying the findings in their manuscript fully available?**

Reviewer #1: Yes

PLOS authors have the option to publish the peer review history of their article (what does this mean?). If published, this will include your full peer review and any attached files.

Reviewer #1: No

---

## [Editor Report · Acceptance letter]

2 Feb 2025

PCOMPBIOL-D-24-00604R1 

An in silico framework for the rational design of vaginal probiotic therapy

Dear Dr Arnold,

I am pleased to inform you that your manuscript has been formally accepted for publication in PLOS Computational Biology. Your manuscript is now with our production department and you will be notified of the publication date in due course.

With kind regards,

Zsofia Freund
